# Seaweed-Based Products and Mushroom β-Glucan as Tomato Plant Immunological Inducers

**DOI:** 10.3390/vaccines8030524

**Published:** 2020-09-13

**Authors:** Paulo César de Melo, Carolina Figueiredo Collela, Tiago Sousa, Diana Pacheco, João Cotas, Ana M. M. Gonçalves, Kiril Bahcevandziev, Leonel Pereira

**Affiliations:** 1Department of Agriculture, Federal University of Lavras/UFLA, Lavras 37200-000, Brazil; pcmelo@ufla.br (P.C.d.M.); collelacarolina@gmail.com (C.F.C.); 2Marine and Environmental Sciences Centre (MARE), Department of Life Sciences, University of Coimbra, 3000-456 Coimbra, Portugal; xtiagosousa97x@gmail.com (T.S.); dianampacheco96@gmail.com (D.P.); jcotas@gmail.com (J.C.); amgoncalves@uc.pt (A.M.M.G.); 3Department of Biology and CESAM, University of Aveiro, 3810-193 Aveiro, Portugal; 4Agricultural College of Coimbra (ESAC/IPC), Research Centre for Natural Resources Environment and Society (CERNAS), Institute of Applied Research (IIA), 3045-601 Coimbra, Portugal; kiril@esac.pt

**Keywords:** *Fusarium oxysporum*, fungicide, *Kappaphycus alvarezii*, tomato

## Abstract

The effects of the abiotic inducers β-glucan, extracted from Shiitake *(Lentinula edodes*), BFIICaB^®^ (*Kappaphycus alvarezii*) and BKPSGII^®^ (*K*. *alvarezii* X *Sargassum* sp.) on tomato plants infected with *Fusarium oxysporum* f. sp. *lycopersici* (FOL) were evaluated through the activity of enzymes related to the induction of resistance at 5 and 10 days after inoculation (DAI). Tomato plants (21 days old, after germination) were inoculated with the pathogen conidia suspension and sprayed with 0.3% aqueous solutions of the inducers. The activities of the enzymes β-1,3-glucanase, peroxidase and phenylalanine ammonia lyase (PAL) were evaluated in fresh tomato leaves collected at 5 and 10 DAI. In all treatments, peroxidase showed the highest enzymatic activity, followed by β-1,3-glucanase and PAL. Between the seaweeds, the inducers extracted from the red alga *Kappaphycus alvarezii* (BFIICaB^®^) promoted the highest enzymatic activity. The exception was BKPSGII^®^ (*K. alvarezii* X *Sargassum* sp.) where the influence of *Sargassum* sp. resulted in higher peroxidase activity (4.48 Δab_600_ mg P^−1^ min^−1^) in the leaves, 10 DAI. Both the red seaweed *K*. *alvarezii* and the brown alga *Sargassum* sp. promoted activities of β-1,3-glucanase, peroxidase and PAL.

## 1. Introduction

Society is raising awareness regarding the impacts of intensive agriculture on the environment. Moreover, the consumer is also concerned with food security, namely the application of synthetic pesticides on crops and their residues on food [1]. In this context, seals that guarantee the proper use of these chemicals and expansion of free food markets have been developed [2]. There is a uniform will and determination to promote the development of eco-friendly pesticides, which is reflected on the 2030 Agenda, under the scope of the Sustainable Development Goal #2, through the promotion of sustainable agricultural practices [3].

Thus, there is a need to develop sustainable and ecologically friendly cropping systems and to replace synthetic chemicals from agricultural production [4,5]. Due to the growing demand for natural products to control pathogen development in plants (i.e., viruses, bacteria, fungi and nematodes), bioprospecting studies seek to evaluate natural resources with such control activities in plants. Marine resources, namely seaweeds and their extracts, contain a wide range of biomolecules that can be effective in controlling plant diseases [5,6]. In fact, algal extracts have widespread applications in the agriculture field, such as fertilizers, plant biostimulants, bio-regulators and growth promoters [6,7]. However, plant immunological responses differ, and the involved mechanisms of action are not yet fully elucidated [8].

Seaweeds and/or their molecules can activate or interact with plants’ latent defense mechanisms, however these mechanisms are only triggered when there is direct plant–pathogen interaction. These mechanisms aim to restrict intruder growth and can lead to induced systemic resistance (ISR) or systemic acquired resistance (SAR), making the plant less susceptible to subsequent pathogen attack [9], which involves the production of elicitors, oxidative bursts and the synthesis of antimicrobial compounds [9,10].

Disease will occur if the pathogen is faster than the induced response, if no elicitors are produced or if suppressors prevent the plant defense reactions [11].

Elicitors are plant molecules/compounds that trigger or stimulate certain defense mechanism in a plant. As a result of the interaction of an elicitor with a receptor of the cell on which it acts, a metabolic stimulus, called a “signal”, is created [12,13]. The elicitor can trigger diverse plant defense mechanisms, such as calcium flux, mitogen-activated protein (MAP) kinase activation, and the production of secondary signals such as reactive oxygen species, nitric oxide, and the phytohormones jasmonic acid, ethylene, and salicylic acid [13]. Salicylic acid is a phenolic compound, present in plants, with elicitor action by inducing the activation of genes that encode pathogenesis-related (PR) proteins and enzymes related to the production of phytoalexins and lignin [14]. The increase in the activity of several enzymes, such as peroxidases, polyphenoloxidases, phenylalanine ammonia lyases, lipoxygenases, β-1,3-glucanases and chitinase, in plant tissues is related to the occurrence of resistance induction [15]. However, PR proteins’ linkage as ISR markers is not yet fully elucidated [9].

In this study, was analyzed the activity of three specific enzymes (β-1,3-glucanase, peroxidase and phenylalanine ammonia lyase), which are triggered by the plant when there is plant–pathogen interaction and are part of the induced resistance mechanism [16,17]. The linear long-chain polymer of β-1,3-glucan is involved in the mechanisms of cellular recognition, in the exogenous plant–pathogen interactions, while β-1,3-glucanase is known as one of the PR proteins and a component of the salicylic acid-dependent pathway, which is related to the plant’s defense mechanism, induced by pathogens or chemical activators [15,18]. This enzyme acts in the degradation of pathogens’ cell wall or membrane, releasing molecules that act as elicitors, in the initial stages of the resistance induction process. The performance of this enzyme is closely linked to the hydrolysis of β-1,3-glucan, one of the major components of the cell wall of many fungi [10,19].

Tomato (*Solanum lycopersicum*) is the most industrialized vegetable crop in Brazil occupying, in 2018, 5 million hectares [20,21]. Tomato culture holds a high economic importance and is affected by several diseases caused by bacteria, fungus, virus or nematodes. Diseases caused by fungi that attack plants from the root system are the most alarming, in particular a vascular wilt [22,23], provoked by a soil-borne pathogen *F*. *oxysporum* f. sp. *lycopersici* (FOL) that attacks tomato crops. The FOL enters in the epidermis of tomato plant root, spreads through the vascular tissue and inhabits the plant xylem vessels, resulting in vessel clogging, and severe water stress symptoms [24] as the plant loses its turgidity. The disease is morphologically identified by wilted plants bearing yellow colored leaves with minimal or absent crop yield [25]. Plants attacked by this pathogen show defoliation from the lower older leaves to the upper younger leaves. Although wilting provoked by FOL is considered dynamic, it was widely accepted that plant response can be a combination of the fungus attack mechanism, an inactivation of the host defense, the growth of the fungal mycelia in the plant xylem and, around the systemic tissue, the production of mycotoxins, such as fuminosin, fusaric acid and tyloses [23]. Fusaric acid is characterized by the phytotoxic effects in tomato leaves, which revealed reduced photosynthesis, leaf wilting and necrosis, enormous lipid peroxidation and intracellular reactive oxygen species and cell death. This acid decreases leaf cell viability, enhancing the fungal pathogenicity [24].

The seaweed extracts are commercially available as biofertilizer [26]. Although they present various bioactive molecules (mainly polysaccharides, phenols, flavonoids, pigments, carotenoids) and important minerals that help the plant to tolerate disease development, the data about the immunomodulatory/immunostimulating activity is scarce [27]. Information regarding chemical characteristics of the algal compounds that can activate the plant immune reaction (e.g., activation of signal pathways regulating inherent defense responses) to any disease attack is insufficient [28]. Various studies have already shown that crude or refined algal extracts can protect plants against pathogens [28,29,30,31,32]. Complex polysaccharides can be found in the algal cell walls and it was already demonstrated that they possess several biological activities, such as eliciting plant defense responses [19,32]. For instance, researchers evaluated the immunomodulatory bioactivity of laminaran, polysaccharide extracted from the brown seaweed *Laminaria digitata*, triggers defense responses in plants, enhancing protection against pathogens [33]. The request for biological control products is increasing, since synthetic fungicides can deposit residues in fruits and vegetables, which make part of a human diet. Thus, the research for products originated from life organisms (bio-products) with anti-pathogen activity is growing [34].

In order to evaluate the resistance in tomato to FOL, the goal of this study was to determine the activity of enzymes related to the induction of resistance (namely, β-1,3-glucanase, peroxidase and phenylalanine ammonia lyase) in tomato plants when induced with the biological fungicide Serenade^®^ (as a positive control) and β-glucan. BFIICaB^®^ (*Kappaphycus alvarezii*) and BKPSGII^®^ (*K. alvarezii* X *Sargassum* sp.) are based on seaweed extracts and used as tomato crop biofertilizers. In this study, the seaweeds’ extracts were also employed to see if they can provoke a plant immune response against the pathogen and to clarify if these seaweed extracts can be applied as a multi-purpose tool (biofertilizer and, if possible, bio fungicides, promoting a plant resistance).

## 2. Materials and Methods

### 2.1. Preparation of the Experiment

The tomato seeds (from the same seed batch) were sown in rigid polypropylene trays, with 162 cells in trapezoidal format with a volume of 31 cm^3^ per cell and dimensions of 35.0 mm length × 35.0 mm wide per cell, in greenhouses, with hanging trays placed on metal benches. The trays were filled with spent mushroom substrate (SMS) from *Agaricus bisporus* and were irrigated three times per day. When plants developed, three adult leaves were transplanted to plastic pots, 1 L volume, containing a commercial substrate Tropstrato^®^ (Vida Verde Indústria e Comércio de Insumos Orgânicos Eireli, Mirim-SP, Brazil).

### 2.2. Preparation of Inoculum and Inoculation

The conidia of FOL isolate (CML1875), provided by the Mycological Collection of Lavras (CML, Lavras, Brazil), were inoculated in a Petri dish containing Potato dextrose agar (PDA) medium (Fungus Genetic Laboratory—Federal University of Lavras, Biology Department, Lavras, Brazil) and incubated for 15 days at the room temperature of 24 °C, in the dark. The suspension was filtered through double gauze, calibrated using a hemocytometer and the inoculum was adjusted to 1 × 10^6^ conidia/mL. Plants were inoculated by drenching the soil around the root zone with the help of a pipette by adding 20 mL of the conidia suspension per plant. Before inoculation, the roots were slightly wounded using a 1 mL syringe fitted with a 25-gauge needle, by inserting the needle 1 cm away from the stem [35]. Root wounding was done to ensure pathogen penetration through roots. Observations were recorded on wilt symptoms for up to 4 weeks.

The assay was conducted in a greenhouse under natural conditions, in the Agriculture Department of the Federal University of Lavras, Brazil. The irrigation was done by the drip technique, every 6 h, with the intention to favor the incidence of symptomatic plants. The drip intervals lasted 5 min. Plant biomass was not evaluated.

A visual methodology of *Fusarium* disease symptoms was used, in order to evaluate disease incidence development.

### 2.3. Inducers

After the conidia of *F. oxysporum* inoculation, the tomato plants were treated with the inducers. The inducers consisted of the biological fungicide Serenade^®^ (as a positive control) (Bayer S.A, São Paulo, Brazil), seaweed extracts of BFIICaB^®^ (*Kappaphycus alvarezii*) and BKPSGII^®^ (*K. alvarezii* and *Sargassum* sp.) (Ceres Tecnologia Agrícola Ltd.a, Lavras-MG, Brazil) and β-glucan, a water-soluble dietary fiber obtained from oats, barley, bacteria, yeast, algae, and mushrooms [36], extracted from Shiitake (*Lentinula edodes*), an edible mushroom, according to Park et al. [37]. Serenade^®^ is a biological fungicide that controls Fusarium development in tomato plants [38,39,40]. The active ingredient in Serenade^®^ is the bacteria *Bacillus subtilis* (strain QST713) [38,41,42,43]. *K. alvarezii* is a red seaweed and belongs to the phylum Rhodophyta (family Solieriaceae), while the brown seaweed *Sargassum* sp. belongs to the class Phaeophyceae (family Sargassaceae).

During the experiment, 20 mL of each treatment was applied on each tomato plant, 5 and 10 DAI. All treatments were done through leaf pulverization (through the total area of the aerial part of the plant), with a commercial pulverizer, at a concentration of 0.3%, except the negative control (treated only with distilled water).

Tomato plants were distributed in completely randomized block design with three repetitions and 15 plants per repetition/treatment, with the total number of plants being 225.

### 2.4. Enzymatic Assays

Samples of 1.0 g of fresh, randomly chosen, leaves from each repetition, harvested 24 h after each treatment (5 DAI and 10 DAI), were macerated in a liquid nitrogen mortar with 1% (*v*/*v*) polyvinylpyrrolidone (PVP) and 1.0 mL of potassium acetate buffer (50 mM/pH 5.0) containing 1 mM EDTA. The extracts were centrifuged at 9000× *g* for 5 min at 4 °C and the supernatant was transferred to Eppendorf tubes and stored at −80 °C [44]. The supernatants were used to evaluate the activity of β-1,3-glucanase, peroxidase and determine the soluble protein content [19]. Each treatment was assayed in triplicate.

#### 2.4.1. β-1,3-Glucanase

To evaluate β-1,3-glucanase activity, 100 μL of the enzyme extract was used; 100 μL of potassium acetate buffer (0.1 M/pH 4.8) and 80 μL of the substrate Carboxymethyl-Curdlan-Remazol Brilliant Blue (CM-Curdlan-RBB) (4 mg/mL). This material was incubated at 40 °C for 90 min, and then 30 μL of hydrochloric acid (2N) was added, being cooled for 10 min on ice and centrifuged 1450× *g* for 10 min at 4 °C. The supernatant was transferred to a new plate and the readings were made on the spectrophotometer at a wavelength of 600 nm [45].

#### 2.4.2. Peroxidase

Peroxidase activity was estimated based on the evaluation of the Δ absorbance provided with guaiacol oxidation in the presence of hydrogen peroxide [44]. For the development of the reaction, 45 μL of guaiacol (0.02 M), 45 μL of hydrogen peroxide (0.38 M) and 100 μL of potassium phosphate buffer (0.2 M/pH 5.8) were pipetted into a spectrophotometric cuvette. Then, 10 μL of the enzyme extract was added and the spectrophotometric reading was performed at a wavelength of 480 nm [19].

#### 2.4.3. Phenylalanine Ammonia Lyase (PAL)

To evaluate PAL activity, 10 μL of the enzyme extract were used, as well as 140 μL of Tris HCl (pH 8.8) and 50 μL of phenylalanine. Readings were performed on a spectrophotometer at a wavelength of 280 nm [19].

### 2.5. Statistical Analysis

A randomized complete block design was used, and the averages were compared using the Tukey test at 5% probability using the SISVAR program [46].

## 3. Results

### 3.1. Tomato Plant Hypersensitivity Reaction to Fusarium oxysporum f. sp. Lycopersici

In tomato plants treated with distilled water (negative control) and inoculated with FOL, tissue necrosis, less foliar growth and, consequently, a loss of photosynthetic activity leading to wilting, was observed (plant on left side, Figure 1a,b) [9,47]. In comparison, on the right side, tomato plants infected and treated with the inducers exhibited more foliar growth (Figure 1a,b).

### 3.2. β-1,3-Glucanase Activity

Five DAI with FOL, the positive control (Serenade^®^), was the treatment that influenced the greatest β-1,3-glucanase activity (3.44 Δab_600_ mg P^−1^ min^−1^) in tomato plants (Figure 2). Comparatively, the other treatments exhibited similar results. For plants treated ten DAI, all treatments presented lower enzymatic activities then plants treated 5 DAI. Compared with the positive control, seaweed samples, the negative control and the β-glucan had lower enzymatic activity. Moreover, BFIICaB^®^ was the sample that exhibited the highest value of this enzyme (1.87 Δab_600_ mg P^−1^ min^−1^), while the β-glucan showed the lowest enzymatic activity (0.5 Δab_600_ mg P^−1^ min^−1^). However, there were significant differences between the treatments after 10 DAI, with the positive control and the seaweed extracts having higher activity.

### 3.3. Peroxidase Activity

Five DAI, the product that provided the highest peroxidase activity in the plant was BFIICaB^®^ (3.59 Δab_480_ mg P^−1^ min^−1^), followed by the negative control (3.43 Δab_480_ mg P^−1^ min^−1^) and β-glucan (3.35 Δab_480_ mg P^−1^ min^−1^), differing substantially from the positive control (2.26 Δab_480_ mg P^−1^ min^−1^) and the BKPSGII^®^ (Figure 3). After the 10th day, there was an overall increase in peroxidase activity, with the negative control and BFIICaB^®^ presenting lower enzymatic activities when compared with the other samples tested.

However, when compared with the positive control, the treatment with β-glucan, the negative control and BKPSGII^®^ (4.48 Δab_600_ mg P^−1^ min^−1^) in the leaves, at 10 DAI, demonstrated higher enzymatic activity and differed significantly from the positive control and BFIICaB^®^.

### 3.4. Phenylalanine Ammonia Lyase (PAL) Activity

After five days, the highest PAL activity (Figure 4) was observed in the positive control and β-glucan, both with 1.74 Δab_480_ mg P^−1^ min^−1^, while the negative control (1.36 Δab_480_ mg P^−1^ min^−1^) exhibited, comparatively, a lower PAL activity. In ten DAI, PAL activity decreased in all treatments. However, the positive and negative controls (0.47 and 0.40 Δab_480_ mg P^−1^ min^−1^, respectively), β-glucan (0.53 Δab_480_ mg P^−1^ min^−1^) and BFIICaB^®^ (0.43 Δab_480_ mg P^−1^ min^−1^) exhibited slight enzymatic activity. BKPSGII^®^ always presented the lowest PAL activity (5 and 10 DAI).

## 4. Discussion

To prevent the infection by the pathogen, the plant uses signaling mechanisms to warn other parts regarding the phytopathogen attack. During this experiment, when seaweed extracts and β-glucan treatments were applied, this was noted on the lower part of the stem, contrary to the results verified when distilled water (negative control) was used, when plant death was imminent.

Jiménez et al. [48] performed in vivo tests and found out that seaweed *Lessonia trabeculata* (Phaeophyceae) reduced both the number and the size of necrotic lesions in tomato leaves after infection with *Botrytis cinerea*, the agent that causes gray rot. Spraying ulvan (sulphated polysaccharide extracted from green macroalgae), to control the fungus *Uromyces appendiculatus* (bean rust) in three different cultivars of carioca beans, promoted an average reduction of 23.8% in the diameter of the rust pustules in bean plants. Although studies confirm the potential of inducing resistance of this polysaccharide against fungi, the mechanisms in which ulvan interferes within the plant system are not yet elucidated [49].

In this assay, it was shown that enzymes β-1,3-glucanase, peroxidase and phenylalanine ammonia lyase (PAL) can trigger three different plant immune responses, which are essential to the control of pathogen attacks and the survival of the plant.

The synthesis and accumulation of pathogenesis-related proteins have been described as playing a key role in the plant defense mechanism (direct antifungal activity) [50,51]. The β-1,3-glucanase is known as lytic enzyme, which acts in the fungi cell wall compounds, mainly chitin and glucan, and also through the ISR in the plant system [50,51]. Consequently, the enzyme, β-1,3-glucanase, has a direct reaction against the cell wall of the wilt fungi *F. oxysporum* and activates the ISR to protect the plant from the fungi attack, mainly inhibiting the fungi growth [52].

The enzyme β-1,3-glucanase is activated quickly when the fungi interacts with the plant to lyse the fungi cell wall. In this period, the seaweed-based products do not appear to be efficient in the rapid response (5 DAI), however in 10 DAI, the β-1,3-glucanase activity is maintained in the treatments assayed.

The β-1,3-glucanase induces the stipe wall extension activity in the fungi [53]. It was also reported that extracts from the brown seaweed *Laminaria digitata* can induce plant systemic resistance. This application resulted in the production of peroxidase and deposition of phenolic compounds causing local cell death of grape downy mildew (*Plasmopara viticola*) [54].

The β-1,3-glucanase activity remained approximately constant in the treatment with BFIICaB^®^ (*K. alvarezii*) and BKPSGII^®^ (*K. alvarezii* X *Sargassum* sp.), inferring that after the activation of the systemic acquired resistance by the seaweed extract, this protein remained active in the plant, providing protection against the pathogen (5 and 10 DAI). It is important to note that this resistance is maybe characterized by the expression of genes related to the production of PR proteins, considered as important factors in the resistance of plants to several classes of phytopathogens [19].

Our results obtained in the treatment with β-glucan extracted from Shiitake (the fungi *Lentinula edodes*), where the activity of β-1,3-glucanase in the tomato plant was observed throughout the evaluated period, corroborate with the works of [23,55,56]. In this way, it is possible to confirm the potential technological application of this polysaccharide extracted from the fruiting bodies of *Lentinula edodes* for the control of FOL in tomato.

If inducers activate peroxidase, which is a plant defense enzyme, as well the phenylalanine ammonia lyase [50], the pathogen can be mitigated. The plant will only have reduced HR where the plant defense mechanism tried to decrease the pathogen attack, by a localized apoptotic effect in the cells adjacent to the pathogen location, an indirect mechanism to control the pathogen propagation. Consequently, the peroxidase also confers higher resistance towards the pathogen, mainly due to the production of hydrogen peroxide [51,57].

The organic fungicide demonstrates an activity against the fungi where the peroxidase activity was lower, among the treatments analyzed. The β-glucan showed some positive effect on enzyme activities. The negative control reacted immediately to the pathogen infection, but it appears that it is not able to maintain the reactive response for a long time, provoking the plant death ten DAI.

Peroxidase’s effect was demonstrated between the 5th and 10th day, inducing systemic acquired protection in tomato plants [58]. Peroxidase activity comes from a sequence of events and signals, starting from a pre-existing isoform, catalyzing the last enzymatic step of lignin biosynthesis, serving as a physical barrier to the pathogen’s penetration, or even interacting with the chitin of the cell walls of many fungi acting together with β-1,3-glucanase [59], resulting in an increase in β-glucan. These two enzymes have a major role in the active defense and in the biosynthesis of various bioactive molecules essential to plant defense mechanisms, such as phenolic compounds or oxygen radical species [50].

The phenylalanine ammonia lyase anti-pathogen activity is involved in the first step of the synthesis of phenylpropanoids/phytoalexins, resulting in the production of compounds such as phytoalexins and lignin, which gives to the plant cell walls greater resistance to the penetration of pathogens [60]. In this case, the pathogen does not die, however, it will be circumscribed to a restricted area where the disease symptoms will appear [23]. PAL is quickly induced during the first stages of the plant–pathogen interaction, decreasing over time.

During the five DAI, PAL activity had a significant increase, that is in agreement with Campos et al. [61]. According to Mercier et al. [62], the red algae *Hypnea musciformis* is the source of the polysaccharide k-carrageenan, which is an elicitor compound that promotes the accumulation of salicylic acid and increases the expression of genes in cells related to defenses against pathogens, and consequently provides the induction of acquired resistance. The red seaweed extracts (*K. alvarezii*) used in this work (BFIICaB^®^ and BKPSGII^®^) promoted the PAL activity in the first five DAI. However, it was not determined which polysaccharide extracted from this alga was decisive in promoting the resistance induction process.

Saravanan et al. [63] analyzed the activity of PAL in banana and beans at six, eight and twelve days after the application of resistance inducers. Saltveit [64] and Chen et al. [65] reported that the highest PAL activity is between 24 to 48 h after the induction. This can be caused due to the linkage between the enzyme and function at the beginning of the lignin biosynthetic route. This can justify why, in our case, PAL activity decreased 10 DAI.

It is possible that the strain used in the study was less aggressive, which is also reflected in the results when positive and negative control treatments were used. *Bacillus subtilis* (the active ingredient in Serenade^®^) showed that it can be used as a prevention treatment, but in the early stage of the FOL development.

Results demonstrated that seaweeds’ extracts mainly promote β-1,3-glucanase and peroxidase activity, involved in indirect mechanisms against the FOL attack. These two enzymes demonstrated higher activity in tomato plants, mainly the peroxidase. Peroxidases exist in isoforms in many plant species, are involved in plant defense, are expressed during cell damage and are present in *Sargassum* sp. [66].

β-glucan and seaweed-based products demonstrated the long-term effect in order to prevent the further FOL penetration, with the activation and maintenance of plant defense mechanisms to control the pathogenic negative impact on the plant. The activation of a defense mechanism influenced the decrease in Fusaric acid production in *F. oxysporum* and thus reduced the severity of wilt symptoms and virulence in the host plant [23].

Thus, our results support that some bio products, designed as biofertilizers, can be used as possible activators of the plant defense mechanisms.

## 5. Conclusions

The extracts based on seaweeds (*Kappaphycus alvarezii* and *K. alvarezii* X *Sargassum* sp.) and β-glucan (extracted from *Lentinula edodes*) presented activities as potential inducers of resistance in tomato plants against Fusarium wilt caused by *Fusarium oxysporum* f. sp. *lycopersici*. The enzymatic activity provoked in tomato plants, when treated with these extracts, can be considered as similar as to ISR or SAR behavior. This behavior was mainly present in plants treated with *Kappaphycus alvarezii*, which can be considered a promising inducer agent and may be used for other crops. This assumption might be addressed in future studies.

## Figures and Tables

**Figure 1 vaccines-08-00524-f001:**
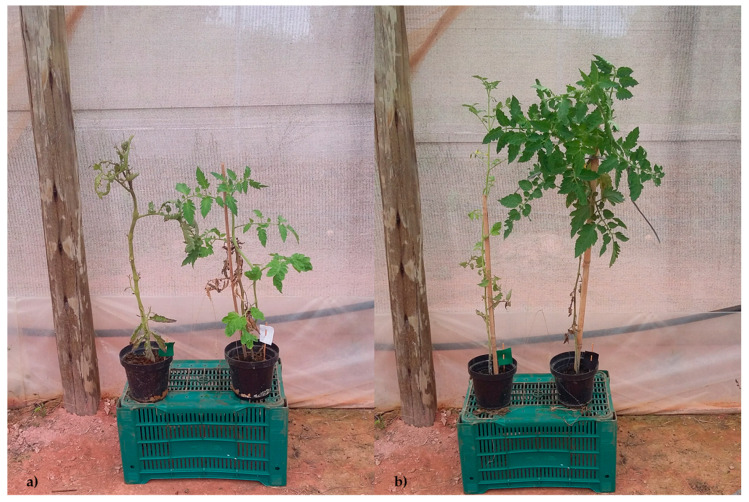
Negative control (left plant) and plant treated with the inducer β-glucan (right plant), 5 (**a**) and 10 (**b**) days after inoculation (DAI).

**Figure 2 vaccines-08-00524-f002:**
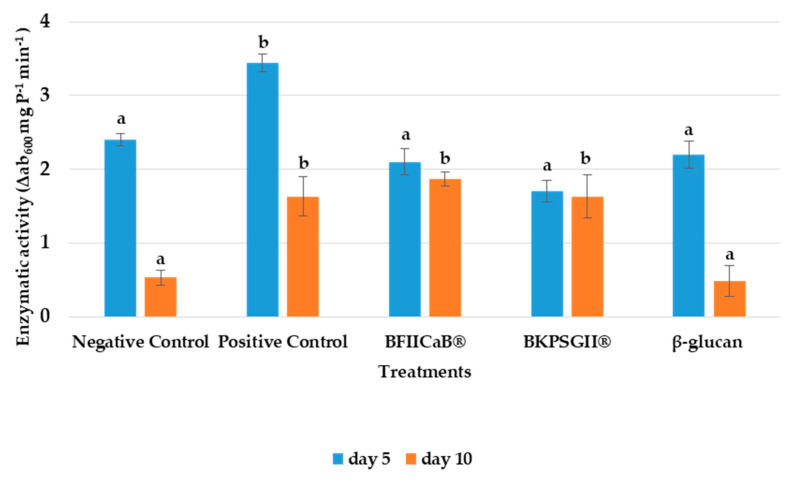
β-1,3-glucanase induced activity in tomato plants inoculated with *Fusarium oxysporum* f. sp. *lycopersici* (FOL) were evaluated 5 and 10 DAI. Plants were treated with distilled water (negative control), Serenade^®^ (positive control), β-glucan, *Kappaphycus alvarezzi* extract (BFIICaB^®^) and BKPSGII^®^. **a**, **b**—Equal letters indicate no significant differences at the *p*-value < 0.05. The statistical analysis was performed separately for each time interval treatment (average ± SD; n = 3).

**Figure 3 vaccines-08-00524-f003:**
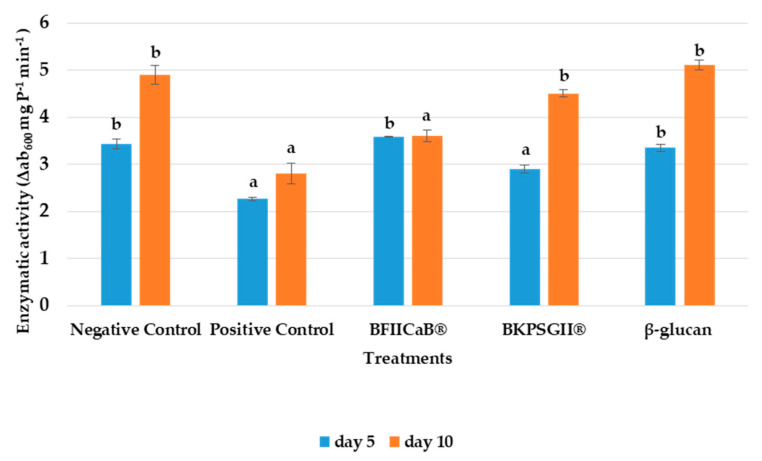
Peroxidase induced activity in tomato plants inoculated with FOL were evaluated 5 and 10 DAI. Plants were treated with distilled water (negative control), Serenade^®^ (positive control), β-glucan, BFIICaB^®^ and BKPSGII^®^. **a**, **b**: Equal letters indicate no significant differences at the *p*-value < 0.05. The statistical analysis was performed separately for each time interval after treatment (average ± SD; n = 3).

**Figure 4 vaccines-08-00524-f004:**
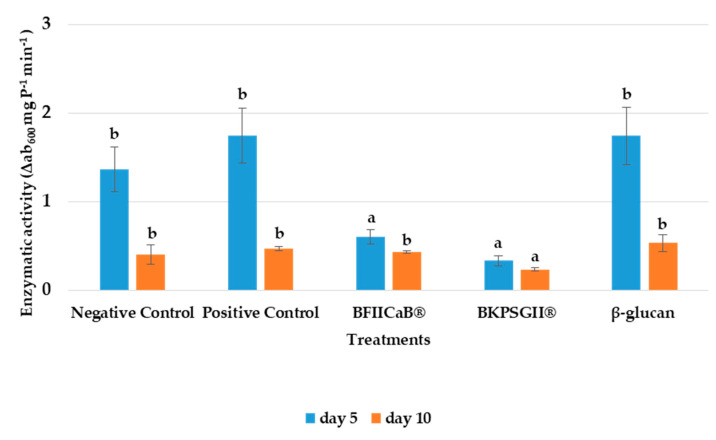
Phenylalanine ammonia lyase (PAL) activity in tomato induced by the different treatments were evaluated 5 and 10 DAI with FOL. Plants were treated with distilled water (negative control), Serenade^®^ (positive control), β-glucan, BFIICaB^®^ and BKPSGII^®^. **a**, **b**: Equal letters indicate no significant differences at the *p*-value < 0.05. The statistical analysis was performed separately for each time interval after treatment (average ± SD; n = 3).

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
