# Peer review of "Seaweed-Based Products and Mushroom β-Glucan as Tomato Plant Immunological Inducers"

_vaccines, 2020, doi:10.3390/vaccines8030524_

Round 1

Reviewer 1 Report

The manuscript entitled "Seaweeds as tomato plant immunological inducers" has novel information.

If possible, please provide the pictorial information at Supplementary data, about the sample preparation (2.1 section).

Please provide a few more details on the data collection, number of treatments repeated for each experiment, etc.

In the results section, did you find any hypersensitivity symptoms in the plants after the infection with conidia spores?

Lines 218-219: What is the peroxidase ratio increased between treatments.

In the Figures, what is the p-value to show the significance? The significant values showing with the alphabets are not different between the treatments in most of the experiments. Can you please write the footnotes? Maybe the 5% p values are too high, is it so?

Results for all the experiments were too short, it will be good to compare the control with each individual treatment.

The quality of the graphs should be improved a bit to the journal standard.

Author Response

Reviewer 1:

Comment 1: The manuscript entitled "Seaweeds as tomato plant immunological inducers" has novel information.

Answer 1: The authors appreciate the reviewer words.

Comment 2: If possible, please provide the pictorial information at Supplementary data, about the sample preparation (2.1 section).

Answer 2: Thank you for pointing this issue. The article was complemented with pictures of the experimental design apparatus in the section 2.1.

Comment 3: Please provide a few more details on the data collection, number of treatments repeated for each experiment, etc.

Answer 3: We added the data in methods section.

Comment 4: In the results section, did you find any hypersensitivity symptoms in the plants after the infection with conidia spores?

Answer 4: Thank you to highlight this issue. More information and images were added in the results, namely in the section 3.1., regarding hypersensitivity symptoms in the tomato plants after the infection with conidia spores.

Comment 5: Lines 218-219: What is the peroxidase ratio increased between treatments.

Answer 5: We added in that section more information and in the discussion.

Comment 6: In the Figures, what is the p-value to show the significance? The significant values showing with the alphabets are not different between the treatments in most of the experiments. Can you please write the footnotes? Maybe the 5% p values are too high, is it so?

Answer 6: The authors improved the design of the graphs and added the standard deviation to the graphs. Also, it was added in the footnote that equal letters indicate no significant differences at the p-value<0.05.

Comment 7: Results for all the experiments were too short, it will be good to compare the control with each individual treatment.

Answer 7: Regarding this issue, the recommendation was followed, and the results were described more in detail in the section 3.

Comment 8: The quality of the graphs should be improved a bit to the journal standard.

Answer 8: Thank you for pointing this issue, the design and quality of the graphs presented were improved.

Reviewer 2 Report

It is well known that seaweed products can induce resistance of tomato plants against fungi. In this paper, authors studied the ability of two commercially available seaweed extracts and ß-glucan as immunological inducers by measuring the activity of three enzymes related in resistance induction 5 and 10 days after treatment. This is a principally acceptable research approach.

However, quality of presentation is poor and the manuscript needs considerable language and editorial improvement.

Some specifics:

  • title is missleading. Authors did not use seaweeds but two commercially available seaweed extracts as inducers
  • Abstract needs rewriting. Enzyme activities were not measured "as results of these treatments" but 5 and 10 days after treatment. The abstract does not show any concrete results
  • line 58: explain PR when first used in the text
  • line 84 and others: correct the grammatical errors (singular/plural and others)
  • line 145 and others: correct misspellings (use of lowercase and uppercase letters etc.)
  • line 148: why is water control called "White"?
  • line 154: Eppendorf is a proper name and should be written in uppercase letter
  • line 162: give g value for centrifugation step
  • line 172 and others: lyase is the correct term
  • first paragraph of Results: no results are presented here. This is part of the Discussion!
  • line 206: "ten days after the DAI" is nonsense
  • Figures 1-3: statistics makes no sense when no SD or SEM values are shown; initial enzyme activities at day 0 are not shown; what means "the statistical analysis was seperated by time"?; 
  • line 241: what means "from Shiitake"? Activity in which plant?
  • line 273: do you mean "differed" from other tretaments.

Author Response

Reviewer 2:

Comment 1: It is well known that seaweed products can induce resistance of tomato plants against fungi. In this paper, authors studied the ability of two commercially available seaweed extracts and ß-glucan as immunological inducers by measuring the activity of three enzymes related in resistance induction 5 and 10 days after treatment. This is a principally acceptable research approach.

Answer 1: The authors appreciate the reviewer words.

Comment 2: However, quality of presentation is poor, and the manuscript needs considerable language and editorial improvement.

Some specifics:

title is misleading. Authors did not use seaweeds but two commercially available seaweed extracts as inducers

Answer 2: The authors revised the title to clarify the work done.

Comment 3: Abstract needs rewriting. Enzyme activities were not measured "as results of these treatments" but 5 and 10 days after treatment. The abstract does not show any concrete results

Answer 3: Thank you for pointing this issue. The authors rewrote the abstract and added more information about the main results.

Comment 4: line 58: explain PR when first used in the text

Answer 4: This issue was rectified.

Comment 5: line 84 and others: correct the grammatical errors (singular/plural and others)

Answer 5: These issues were rectified.

Comment 6: line 145 and others: correct misspellings (use of lowercase and uppercase letters etc.)

Answer 6: These issues were rectified.

Comment 7: line 148: why is water control called "White"?

Answer 7: Regarding this issue, the authors decided to denominate the ‘’white’’ as the negative control and the product Serenade as the positive control, because it is more scientifically correct.

Comment 8: line 154: Eppendorf is a proper name and should be written in uppercase letter

Answer 8: This issue was rectified.

Comment 9: line 162: give g value for centrifugation step

Answer 9: Thank you for pointing this issue. The information required was added to the text (“1450 g for 10 min at 4 °C”).

Comment 10: line 172 and others: lyase is the correct term

Answer 10: This issue was corrected.

Comment 11: first paragraph of Results: no results are presented here. This is part of the Discussion!

Answer 11: Thank you for your careful observation. In this context, the first paragraph was removed from this section and coupled to the discussion.

Comment 12: line 206: "ten days after the DAI" is nonsense

Answer 12: This issue was corrected

Comment 13: Figures 1-3: statistics makes no sense when no SD or SEM values are shown; initial enzyme activities at day 0 are not shown; what means "the statistical analysis was separated by time"?;

Answer 13: The authors improved the design of the graphs and added the standard deviation to the graphs.

Enzyme activities assays were not performed at day 0.

It means that the statistical analysis was performed separately for each time interval after treatment. The authors rewrote the sentence in order to clarify the message.

Comment 14: line 241: what means "from Shiitake"? Activity in which plant?

Answer 14: This sentence is referring to the increase of the enzymatic activity in tomato plant, when applied β-glucan extracted from the fungus Lentinula edodes (also known as Shittake mushroom). Thank you for your careful observation, the authors rewrote the sentence, in order to be more explicit.

Comment 15: line 273: do you mean "differed" from other treatments.

Answer 15: Thank you for the observation, this issue was tackled.

Reviewer 3 Report

This article lacks of originality. The systemic  acquired resistance induction mechanisms in plants investigated by the Authors are already known so results do not add anythink new to the subject. As a matter of fact the resistance inducers they tested are commercial products largely used in agriculture. The experimental design is not appropriate: I did not find any mention of how disease symptoms on artificially inoculated plants correlated with the enzymatic activity. Inoculation methods are not described in detail and disease assesment methods are not reported at all. All section 3.1 is speculative and do not contain experimental data so it is not appropriate to include it into the section Results. Moreover results are not congruent as in many cases values in plants treated with resistance inducers  do not differ significantly from the control. No explanation is given of this. Moreover the  enzymatic activity varied randomly at different time intervals after the treatment with inducer substances. It follows that  the discussion  is not consequential. 

The English style must be substantial improved and the scientific terminology is not always adequate.

In general I am wondering if the subject of induced resistance in plants may be compared to the action of vaccines in human and animal medicine which is based on immunity. Is Vaccines the right jouranl for this kind of articles? I think that Authors should discuss this aspect and compare different mechanisms.

References should be checked carefully.

For other minor suggestions see notes in the text (attached file)

Author Response

Reviewer 3:

Comment 1: This article lacks originality. The systemic acquired resistance induction mechanisms in plants investigated by the Authors are already known so results do not add anythink new to the subject. As a matter of fact, the resistance inducers they tested are commercial products largely used in agriculture. The experimental design is not appropriate: I did not find any mention of how disease symptoms on artificially inoculated plants correlated with the enzymatic activity. Inoculation methods are not described in detail and disease assessment methods are not reported at all. All section 3.1 is speculative and do not contain experimental data so it is not appropriate to include it into the section Results. Moreover, results are not congruent as in many cases values in plants treated with resistance inducers do not differ significantly from the control. No explanation is given of this. Moreover, the enzymatic activity varied randomly at different time intervals after the treatment with inducer substances. It follows that the discussion is not consequential.

Answer 1: The authors thank the reviewer for your words, which were helpful to improve our manuscript. We disagree in part to the lack of novelty, because we tested commercial solution of fertilizer that are supposed to not have bioactivities associated. However, in our study we observed the effect of commercial product against one of the main tomato pathogens that have huge impact in the crops nowadays, so it is an area where is needed to research to protect the crop from the pathogens. And also, this type of correlation between three inducers is not common in the bibliography analyzed, so we give an insight of how the ISR are activated and how the tomato plant benefits of each treatment (in the discussion, we added a clarification of this part to be clearer).

The manuscript was revised with addition of more information and data that the reviewer recommends to us. We also trimmed some parts in the results and discussion.

Comment 2: The English style must be substantial improved, and the scientific terminology is not always adequate.

Answer 2: The authors revised the manuscript.

Comment 3: In general, I am wondering if the subject of induced resistance in plants may be compared to the action of vaccines in human and animal medicine which is based on immunity. Is Vaccines the right journal for this kind of articles? I think that Authors should discuss this aspect and compare different mechanisms.

Answer 3: The treatment used is equip arable to a vaccine, where the seaweed extract does not act directly against the pathogen, but help the inherent plant immune mechanism to fight against the infection pathogen. So, it is considered an immune inducer as a vaccine (which activates the inherent latent plant defense mechanism against a pathogen). And this article is for a special issue about the immune mechanism of the plants.

Comment 4: References should be checked carefully.

Answer 4: The authors revised the bibliography.

Comment 5: For other minor suggestions see notes in the text (attached file)

Answer 5: Thank you for your careful observations. The notes of the attached file were considered, and the incorporations and modifications were done along the text.

Round 2

Reviewer 2 Report

The manuscript has been improved, but still needs some editorial and language corrections.

  • Fig. 1 does not provide significant information and should be deleted
  • line 166: 5.6 and not 5,6
  • line 169: subheading in lowercase letters
  • line 177: all words in the subheading in italics
  • line 205 and others: all figure legends should show the number of data (n) and kind of data variation (SD or SEM)
  • lines 237ff: this sentence is nonsense
  • line 340: and can be used for others crops
  • References still do not follow Authors' Instructions (correct acronyms for journal titles; paper titles in lowercase letters etc.)

Author Response

Reviewer 2:

Comment 1: The manuscript has been improved, but still needs some editorial and language corrections.

Answer 1: Thank you for your considerations and helpful insights. The English editing was revised by a professional reviewer.

Comment 2: Fig. 1 does not provide significant information and should be deleted

Answer 2:  Figure 1 was deleted.

Comment 3: line 166: 5.6 and not 5,6

Answer 3: Thank you for pointing this issue. The authors corrected the information.

Comment 4: line 169: subheading in lowercase letters

Answer 4: Thank you for pointing this issue. The authors corrected the text formatting.

Comment 5: line 177: all words in the subheading in italics

Answer 5: Thank you for pointing this issue. The authors corrected the text formatting.

Comment 6: line 205 and others: all figure legends should show the number of data (n) and kind of data variation (SD or SEM)

Answer 6: Thank you for pointing this issue. The authors added this information in all figure caption.

Comment 7: lines 237ff: this sentence is nonsense

Answer 7: Thank you for pointing this issue. The authors rewrote the sentence.

Comment 8: line 340: and can be used for others crops

Answer 8: Thank you for pointing this issue. The authors rewrote the sentence.

Comment 9: References still do not follow Authors' Instructions (correct acronyms for journal titles; paper titles in lowercase letters etc.)

Answer 9: We revised the reference following the Authors’ instructions.

Reviewer 3 Report

I very much appreciated the efforts made by the Authors to improve the quality of the article. However I am afraid I have to confirm my negative evaluation. I reported some minor comments in the text (see notes in the attached file).  Major criticisms are as it follows: 

  • Both the English style and the technical language remain very poor and not adequate  for an international scientific journal. I must confess it was very hard for me to understand what the Author mean.
  • Fusarium wilt of tomato is a typical vascular disease and the general resistance mechanisms of plants reported by the Authors may not operate in this pathosystem.
  • The description of disease symptoms on inoculated plants is  a little bit elementary: how did the Authors detect the production of conidia in the vascular systems? What they mean for reduction of leaf growth? 
  • The description of materials and methods is not adequate: a) the Authors did not specify if the tomato plants were treated before or after the inoculation which is crucial in the induction of resistance to plant pathogens; b) the inoculation method is not clearly described, i.e. did the Authors injected the conidial suspension into the roots?; c) growth conditions for test plants were not reported.
  • The experimental design presents some substantial flaws. I do not understand e.g.  why the Authors used the commercial product Serenade (Bayer) as a reference (they indicate this treatment as positive control); Serenade is not reported to be effective against vascular diseases, it has been registered by the manufacturer for many fungal and bacterial plant  pathogens, including Botrytis spp., Pseudomonas spp., etc. but as far as I know it has not been registered and has not be reported to be effective against vascular diseases caused by Fusarium spp. So it seems to me quite strange it was used as a control and resulted to be effective in controlling this kind of disease (this  would be very interesting and would deserve to be commented as Serenade is a commercial product used worldwide).
  • Section "Results": statements are not consistent with the Figures ,  terms like "the best product" are not suitable for a scientific article, if two results are not statistically different they  can not be considered different (one "the best" the other one  "intermediate"). 
  • The Discussion is not consequential with the results and the Authors did not consider that the pathosystem they studied is a vascular disease where e.g. toxins produced by the pathogen are involved in the infection process and detoxification is a possible resistance mechanism. In general it was very hard to follow the Author's considerations and in particular  the relationships between the theoretical basis of induced resistance (immunity) in plants and the experimental results presented in this article.
  • There is a discrepancy  between the Author's statement that the algal extracts they tested are  experimental products (see Lanes 137-138) and the adjective "commercial" they used throughout the text

Author Response

Reviewer 3:

Comment 1:  I very much appreciated the efforts made by the Authors to improve the quality of the article. However, I am afraid I have to confirm my negative evaluation.

Answer 1: We are thankful for the time and effort that the reviewer has dedicated to providing valuable feedback on our manuscript. We hope that with the changes and the information added by the authors according to the suggestions made, improved the manuscript.

Comment 2: I reported some minor comments in the text (see notes in the attached file).  Major criticisms are as it follows:

Answer 2: We revised all the manuscript; however, it was not sent an attached file of your review.

Comment 3: Both the English style and the technical language remain very poor and not adequate for an international scientific journal. I must confess it was very hard for me to understand what the Author mean.

Answer 3: Thank you for pointing this issue. A professional reviewer revised the English editing and technical language.

Comment 4: Fusarium wilt of tomato is a typical vascular disease and the general resistance mechanisms of plants reported by the Authors may not operate in this pathosystem. –

Answer 4: Thank you to indicate this observation. The authors added more information in the introduction section, in order to justify the enzyme selection and supporting it with bibliography.

Comment 5: The description of disease symptoms on inoculated plants is a little bit elementary: how did the Authors detect the production of conidia in the vascular systems?
What they mean for reduction of leaf growth?

Answer 5: More information was added in the section “2.2 Preparation of inoculum and inoculation. A visual methodology of Fusarium disease symptoms was used, in order to evaluate disease incidence development. As it is shown in Figure 1a) and 1b), the left plant (negative control) was characterized by tissue necrosis, lower foliar growth and consequently a loss of photosynthetic activity leading to wilting.”

Comment 6: The description of materials and methods is not adequate: a) the Authors did not specify if the tomato plants were treated before or after the inoculation which is crucial in the induction of resistance to plant pathogens; b) the inoculation method is not clearly described, i.e. did the Authors injected the conidial suspension into the roots?; c) growth conditions for test plants were not reported.

Answer 6: 5 and 10 DAI means after the inoculation. The section of materials and methods was revised, and it was added more detailed information about the assay.

Comment 7: The experimental design presents some substantial flaws. I do not understand e.g. why the Authors used the commercial product Serenade (Bayer) as a reference (they indicate this treatment as positive control); Serenade is not reported to be effective against vascular diseases, it has been registered by the manufacturer for many fungal and bacterial plant pathogens, including Botrytis spp., Pseudomonas spp., etc. but as far as I know it has not been registered and has not be reported to be effective against vascular diseases caused by Fusarium spp. So it seems to me quite strange it was used as a control and resulted to be effective in controlling this kind of disease (this would be very interesting and would deserve to be commented as Serenade is a commercial product used worldwide).

Answer 7: It is justified in the section 2.3 Inducers; the Serenade is an antifungal agent against Fusarium in tomatoes supported by the Bayer and also, the article of Cucu, M.A.; Gilardi, G.; Pugliese, M.; Gullino, M.L.; Garibaldi, A. An assessment of the modulation of the population dynamics of pathogenic Fusarium oxysporum f. sp. lycopersici in the tomato rhizosphere by means of the application of Bacillus subtilis QST 713, Trichoderma sp. TW2 and two composts. Biol. Control 2020, 142, 104158, doi:10.1016/j.biocontrol.2019.104158.

Comment 8: Section "Results": statements are not consistent with the Figures , terms like "the best product" are not suitable for a scientific article, if two results are not statistically different they can not be considered different (one "the best" the other one "intermediate").

Answer 8: We revised the section and we trimmed the text in the section.

Comment 9: The Discussion is not consequential with the results and the Authors did not consider that the pathosystem they studied is a vascular disease where e.g. toxins produced by the pathogen are involved in the infection process and detoxification is a possible resistance mechanism. In general it was very hard to follow the Author's considerations and in particular the relationships between the theoretical basis of induced resistance (immunity) in plants and the experimental results presented in this article.

Answer 9: We redid the discussion section, and full revised the section to be easier to follow and understand. It was considered as Reviewer suggested. We did not analyze the resistance to FOL, but the effect of different inducers to possible plant defense enzymatic activity. That is why we did not evaluate the disease incidence or divide the plant response to FOL in different classes (0 to 5) or even observe conidia development in the plant tissue. Plant was considered as infected (or no infected). Regarding the toxins, they are mentioned, in Introduction and Discussion, but they were also not our aim of study.

Comment 10: There is a discrepancy between the Author's statement that the algal extracts they tested are experimental products (see Lanes 137-138) and the adjective "commercial" they used throughout the text

Answer: 10: Corrected in section 2.3.

Round 3

Reviewer 3 Report

The article has been substantially improved after revisions and all criticisms of the reviewers have been properly addressed. I think that the paper can be accepted in this form. I reported just few very minor corrections in the text (see attached file)

Author Response

Reviewer 3:

Comment 1: The article has been substantially improved after revisions and all criticisms of the reviewers have been properly addressed. I think that the paper can be accepted in this form. I reported just few very minor corrections in the text (see attached file).

Answer 1: The authors would like to acknowledge the comments and revisions that allowed the improvement of the manuscript. The comments on the attached file were considered and corrected.